# The BigScience ROOTS Corpus:
# A 1.6TB Composite Multilingual Dataset

Hugo Laurençon[1*]    Lucile Saulnier[1*]    Thomas Wang[1*]    Christopher Akiki[2*]
Albert Villanova del Moral[1*]    Teven Le Scao[1*]

Leandro von Werra[1]    Chenghao Mou[3]    Eduardo González Ponferrada[4]    Huu Nguyen[5]
Jörg Frohberg [32]    Mario Šaško [1]    Quentin Lhoest [1]

Angelina McMillan-Major[1,6]    Gérard Dupont[7]    Stella Biderman[8,9]    Anna Rogers[10]
Loubna Ben allal[1]    Francesco De Toni[11]    Giada Pistilli[1]    Olivier Nguyen [28]
Somaieh Nikpoor[12]    Maraim Masoud[13]    Pierre Colombo[14]    Javier de la Rosa[15]
Paulo Villegas[16]    Tristan Thrush[1]    Shayne Longpre[17]    Sebastian Nagel[19]    Leon Weber [20]    Manuel Romero Muñoz [21]    Jian Zhu [22]    Daniel van Strien [23]    Zaid Alyafeai [24]
Khalid Almubarak [25]    Vu Minh Chien [26]    Itziar Gonzalez-Dios [27]    Aitor Soroa [27]
Kyle Lo [29]    Manan Dey [30]    Pedro Ortiz Suarez [31]    Aaron Gokaslan [18]    Shamik Bose[3]
David Ifeoluwa Adelani[33]    Long Phan[34]    Hieu Tran[34]    Ian Yu[35]    Suhas Pai[36]
Jenny Chim[37]

Violette Lepercq[1]    Suzana Ilić[1]    Margaret Mitchell[1]    Sasha Luccioni[1]    Yacine Jernite[1]

[1]Hugging Face    [2]Leipzig University    [3]Independent Researcher    [4]Ferrum Health
[5]Ontocord.ai    [6]University of Washington    [7]Mavenoid    [8]EleutherAI    [9]Booz Allen Hamilton
[10]University of Copenhagen    [11]University of Western Australia    [12]CAIDP
[13]Independent Researcher    [14]CentraleSupélec    [15]National Library of Norway
[16]Telefonica I+D    [17]MIT    [18]Cornell University    [19]Common Crawl
[20]Humboldt-Universität zu Berlin and Max Delbrück Center for Molecular Medicine    [21]Narrativa
[22]University of Michigan, Ann Arbor    [23]British Library
[24]King Fahd University of Petroleum and Minerals
[25]Prince Sattam bin Abdulaziz University (PSAU)    [26]DETOMO Inc.
[27]HiTZ Center, University of the Basque Country (UPV/EHU)    [28]ServiceNow
[29]Allen Institute for AI    [30]SAP    [31]Mannheim University    [32]Apergo.ai    [33]Saarland University
[34]VietAI Research    [35]Aggregate Intellect    [36]Bedrock AI    [37]Queen Mary University of London

* Equal contributions

## Abstract

As language models grow ever larger, the need for large-scale high-quality text datasets has never been more pressing, especially in multilingual settings. The BigScience workshop, a 1-year international and multidisciplinary initiative, was formed with the goal of researching and training large language models as a values-driven undertaking, putting issues of ethics, harm, and governance in the foreground. This paper documents the data creation and curation efforts undertaken by BigScience to assemble the Responsible Open-science Open-collaboration Text Sources (**ROOTS**) corpus, a 1.6TB dataset spanning 59 languages that was used to train the 176-billion-parameter BigScience Large Open-science Open-access Multilingual (**BLOOM**)(BigScience Workshop, 2022) language model. We further release a large initial subset of the corpus and analyses thereof, and hope to empower large-scale monolingual and multilingual modeling projects with both the data and the processing tools, as well as stimulate research around this large multilingual corpus.

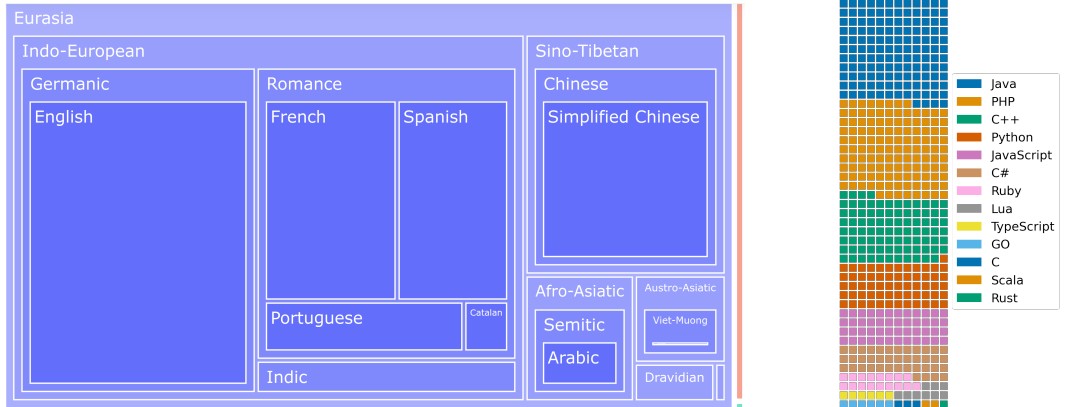

Figure 1: Overview of ROOTS. Left: A treemap of natural language representation in number of bytes by language family. The bulk of the graph is overwhelmed by the 1321.89 GB allotted to Eurasia. The orange rectangle corresponds to the 18GB of Indonesian, the sole representative of the Papunesia macroarea, and the green rectangle to the 0.4GB of the Africa linguistic macroarea. Right: A waffle plot of the distribution of programming languages by number of files. One square corresponds approximately to 30,000 files.

# 1 Introduction

BigScience[1] started in May 2021 as a one-year long open collaborative research initiative that gathered over a thousand participants around the world to study large language models (LLM). One of the founding goals of BigScience was to train an open-access, massively multilingual LLM, comparable in scale to GPT-3 (Brown et al., 2020) yet trained on a better documented and more representative multilingual dataset. The overall BigScience workshop was designed as a collaborative (Caselli et al., 2021; Bondi et al., 2021) and value-driven (Birhane et al., 2021) endeavor. Throughout the process of building this corpus we engaged in simultaneous investigation of ethical (Talat et al., 2022), sociopolitical (McMillan-Major et al., 2022), and data governance issues (Jernite et al., 2022) with the explicit goal of doing good for and by the people whose data we collected.

Sourcing and building the dataset was organized around four working groups: **Data Governance** which helped define the project's values and design our approach to data usage and release in an international context, **Data Sourcing and Preparation** which was tasked with overseeing data collection, curation efforts, and **Privacy** for privacy risks and sanitizing the dataset, **Legal Scholarship** which helped define the multi-jurisdiction legal context in which the entire workshop was to operate, and we discuss practical implications throughout the paper where appropriate. An overview of the BigScience Corpus is provided in figure 1.

The goal of the current paper is twofold: (1) we present a preliminary gated, subject to committing to the BigScience ethical charter[2], release of a large subset of ROOTS[3] (2) we release the numerous data tools[4] that were developed along the way and enabled us to curate, source, clean and inspect all 498 constituent datasets that come together to constitute ROOTS. This includes a preliminary results of the analyses that are currently being developed to study the corpus.

## 1.1 Outline of the Paper

The remainder of this paper details our approach to curating a web-scale dataset covering 59 languages, 46 natural languages and 13 programming languages — the language choice was chiefly driven by the communities who participated in the effort given the importance we placed on language expertise. Our final corpus is made up of two main components: 62% of the text comes from a community-selected and documented list of language data sources and its collection process is described in section 2, and

---

[1]https://bigscience.huggingface.co/
[2]https://hf.co/spaces/bigscience/ethical-charter
[3]https://hf.co/bigscience-data
[4]https://github.com/bigscience-workshop/data-preparation

38% consists of text extracted from a pre-processed web crawl, OSCAR (Ortiz Suárez et al. (2020)), filtered with the help of native speakers, which is described in section 3.

## 1.2 Related Work

**Large Language Models and Large Text Corpora** The current dominant paradigm in natural language processing relies heavily on pre-trained models: large language models that can then be fine-tuned on a downstream task (Howard and Ruder, 2018; Devlin et al., 2018) or even used as-is without additional data (Radford et al., 2019; Brown et al., 2020). In this paradigm, performance is directly correlated on both the model size and the dataset size and quality (Kaplan et al., 2020), with recent models trained on up to 1.4 trillion tokens (Hoffmann et al., 2022) and dataset creation pipelines representing a significant part of large language model projects. Most such datasets, however, are not released, hindering further research. Exceptions include the Pile (Gao et al., 2020), a curated corpus of datasets for language modeling that has become widely used for training state-of-the-art English-language models (Lieber et al., 2021; Smith et al., 2022; Black et al., 2022; Zhang et al., 2022), and C4 and mC4 (Raffel et al., 2020; Xue et al., 2020), which have powered the T5 family of models; CC100 (Conneau et al., 2020) which has seen heavy use for multilingual modeling; and OSCAR (Ortiz Suárez et al., 2019), which has enabled monolingual non-English models.

**Tooling, Visualization, and Replication** Upstream from the finalized training datasets is the issue of processing methods and pipelines: both the operations that the datasets go through and the engineering effort required to apply them at terabyte scales. Existing work tends to fall on a spectrum from no details at all (Brown et al., 2020) to detailed filtering instructions, with (Raffel et al., 2020) or without the dataset release (Rae et al., 2021) to detailed filtering instructions with the accompanying code (Gao et al., 2020; Conneau et al., 2020; Ortiz Suárez et al., 2019). Even when the code is released, it tends to be built and tailored for the project's purpose. Consequently, large projects that do not re-use an existing dataset outright usually build their own pipeline rather than re-use an existing one on new data. However, data tools that were built and packaged in order to be used for other projects exist, such as OSCAR's Ungoliant and Goclassy (Abadji et al., 2021; Ortiz Suárez et al., 2019), which provides a distributed Common Crawl processing pipeline; CCNet (Wenzek et al., 2020), built for quality filtering of multilingual Common Crawl dumps; and OpenWebText (Gokaslan and Cohen, 2019), enabling Reddit dump processing.

**Documenting Textual Corpora in NLP** An inspiration for our work is a recent emphasis on a more in-depth documentation of what is included and what is not in the corpora used for training NLP models . The most notable example of this is the Pile, for which the authors themselves analyze and document a variety of syntactic and semantic properties of the dataset including structural statistics (n-gram counts, language, document sizes), topical distributions across its components, social bias and sentiment co-occurrence, pejorative content, and information about licensing and authorial consent, in addition to releasing a datasheet (Biderman et al., 2022). Other LM pre-training datasets that have been documented and analyzed include C4 (Dodge et al., 2021; Luccioni and Viviano, 2021; Kreutzer et al., 2022), OSCAR (Kreutzer et al., 2022) and BookCorpus (Bandy and Vincent, 2021) . While this kind of documentation is far from standard practice, it is becoming increasingly common given recent calls for better documentation (Rogers, 2021; Bender et al., 2021) as well as empirical studies on data memorization in language models (Carlini et al., 2019, 2022).

## 2   (Crowd) Sourcing a Language Resource Catalogue

The first part of our corpus, accounting for 62% of the final dataset size (in bytes), was made up of a collection of monolingual and multilingual language resources that were selected and documented collaboratively through various efforts of the BigScience Data Sourcing working group. The first such effort consisted in creating a tool to support metadata collection through open submissions, called the BigScience Catalogue and running a series of hackathons in collaboration with locally-focused ML and NLP communities such as Masakhane, Machine Learning Tokyo and LatinX in AI where participants could add and document entries for their languages to the catalogue (McMillan-Major et al., 2022). This yielded a set of 252 sources, including at least 21 per considered language category. We focused on metadata collection as a way to support selection of the sources for the final dataset and documentation of the final dataset. In parallel, working group participants gathered additional

Arabic language resources in the Masader repository (Alyafeai et al., 2021), and proposed a list of websites of interest to increase the geographical diversity of our English, Spanish, and Chinese language data. Finally, in order to explicitly test large language models' ability to handle computer code along with natural language, we selected code data available on GitHub and StackExchange.

## 2.1 Obtaining Data from the Identified Resources

**Gathering Identified Datasets and Collections.** First, we leveraged the BigScience Catalogue and the Masader repository to start obtaining text from identified sources, which included both existing NLP datasets and collections of documents of various compositions. Given the diversity of sources, hosting methods, data custodians, and formats, collecting this text required a collaborative effort. To that end, we established a 2-phase approach: first, collect as many data sources as possible in an easily accessible location; second, map all of them to a common format to ease further processing.

In the first phase, we organized an open hackathon to start gathering identified sources on the Hugging Face Datasets hub (Lhoest et al., 2021), in a dedicated organization[5] (in order to manage access controls). In the second phase, the collected datasets were furthered processed via (1) *Language segmentation*, whereby data sources were split using metadata for each covered language in order to obtain monolingual datasets, and the use of (2) *Uniform interface* whereby a document consisted of two fields: "text" for the actual text content, and "meta" with a JSON representation of metadata for a given document, containing sufficient information to trace documents back to their original sources.

**Pseudo-Crawled Data.** Of the various categories of language resources identified through the data sourcing effort, websites stood out as one that required a particular effort and dedicated pipeline. We decided to design such a pipeline based on "pseudo-crawling": that is, rather than crawling the websites ourselves, we retrieved pages corresponding to the target domain names from 18 snapshots archived by Common Crawl in 2020 and 2021 in Web ARChive (WARC) format (Mohr et al., 2008). These domain names came from two main sources: the homepage field in the metadata of the 252 above-mentioned catalogue entries when available (192 in total), and the 456 websites proposed by participants asynchronously to improve the geographical diversity of our language sources; which yielded a total of 614 unique domain names after deduplication.

We collected URLs contained within those domains using the Common Crawl index. The index provides metadata for every document including the page URL, WARC filename and record offsets, fetch status, content MIME type, etc. We ran a query matching all documents that share the domain name with a seed using Amazon Athena on Common Crawl's columnar index[6]. 48 of the 614 initial seed domain names had no matches in the index and were therefore left out. Once we obtained the document metadata, we fetched the WARC records using HTTP range requests with the start and end byte offsets. Since HTML web pages constitute the largest portion of pages contained in the Common Crawl dumps, we decided to only extract text from HTML pages. Documents in other formats were filtered out, ie XML, PDF, etc. 27 domain names were additionally removed from the list at this stage as we had not retrieved any HTML pages for them.

To extract the text from the HTML pages, we first minified the HTML code. Minification is the removal of unnecessary characters from the source code of a website. Inspired by Aghajanyan et al. (2022), we removed from the DOM-HTML all the sub-trees contained in a *<script>*, *<style>*, *<header>*, *<iframe>*, *<footer>* and *<form>* tag as well as all the sub-trees associated with a *<body>*, *<div>*, *<p>*, *<section>*, *<table>*, *<ul>*, *<ol>* or *<dl>* tag whose textual content was less than 64 characters long. The text was then extracted from the nodes of this new DOM-HTML. While concatenating the text extracted, we applied a set of rules to reconstruct the structure of the text without its HTML code, inspired by what Common Crawl does to extract its WET files (Appendix B.1). The overall procedure enabled us to obtain text datasets for 539 domain names.

**GitHub Code.** We collected a code dataset from BigQuery[7] using the same language selection as AlphaCode (Li et al., 2022). The dataset was then deduplicated of exact matches and filtered for source files with between 100 and 200,000 characters, between 15-65% alphabetic characters, a max

---

[5]https://hf.co/bigscience-catalogue-data
[6]https://commoncrawl.org/2018/03/index-to-warc-files-and-urls-in-columnar-format/
[7]"GitHub on BigQuery: Analyze all the open source code"

line length of 20-1000 characters, and a token length standard deviation of more than 3. Due to a bug in the pre-processing pipeline the dataset was also filtered for GPL licenses only.

**Merging and Deduplicating Sources.** After gathering and processing language data via the three pipelines outlined above, we took a final step to manually inspect, deduplicate, and make a further selection of the sources. First, we addressed dataset overlap we found by looking through our sources. For example: *OpenITI* was present in both its raw form as well as a processed version. Consensus was reached to choose the latter version. Non-trivial datasets overlap included *s2orc* (Lo et al., 2020), *Arxiv* (Clement et al., 2019) and the *PubMed Central* subset of the Pile (Gao et al., 2020). We also performed cross-pipeline dataset deduplication, removing the pseudo-crawled Wikipedia and GitHub in favor of their other versions. We also removed datasets that we found had a high incidence of documents that were not fully in natural language (e.g. unexpected instances of SEO, HTML tags etc...), as well as very small datasets in the higher-resourced languages. Finally, pseudo-crawled sources were further processed to remove menus (with a heuristic consisting of removing lines that occurred in more than 1% of pages in a given domain) and pages that had a high incidence of character ngram repetition, low language identification confidence, or low proportion of closed class words (see Section 3). We then removed entire domains whose size was less than 2MB after this step, yielding 147 pseudo-crawl-based datasets, and a total of 517 datasets including all three pipelines.

## 2.2 Processing Pipeline for Quality Improvement on Crowdsourced Datasets

Once a text field was obtained, we attempted to improve the quality of that text. In the specific case of text extraction from HTML, we observe that not all text are relevant (menus, advertisements, repeated text on each page etc ...). In order to remove noisy data from our dataset, we applied a processing pipeline for each dataset consisting of a sequence of functions.

Functions were categorised as *document-scoped* or *dataset-scoped* functions. *Document-scoped* functions are operations that modify a document independently of other documents and *dataset-scoped* functions are operations that take into account the whole dataset. Orthogonal to this scope, functions were also separated into *cleaning* and *filtering* functions. *Cleaning functions* aim to remove text considered not part of the main document. Document-scoped cleaning functions can for example target leftover HTML tags. On the other end, dataset-scoped cleaning functions need the whole dataset to calculate a heuristic to determine how to modify each document. For instance, advertisements vary across datasets, making it harder to define a dataset-agnostic classifier for advertisement. Instead, we can index all the lines in a dataset and identify repeated lines on multiple pages as likely advertisements. An example is displayed in Appendix B.2. *Filtering functions* aim at removing an entire document from the corpus. The reasons for choosing to remove a document completely are diverse: it may be because the document is considered to be of too poor quality, to be too complex to automatically fix or too similar to other examples already present in the corpus. In the latter case, we speak of deduplication. Deduplication of a document is dependent on whether an equivalent document already exists somewhere else in the dataset and is thus necessarily a dataset-scope function. The notion of equivalent documents has been explored by Lee et al. (2022). In this case we provide deduplication via metadata (urls, normalised urls) and via text (exact string matching). An exhaustive list of functions is available in B.3.

As datasets came from heterogeneous sources with different properties, each needs its own set of processing functions to correspond to our definition of natural language documents. In order to support participants in deciding what functions to apply to which, we built and released a *streamlit*-based visualization tool (figure 2 helps understand the impact of each function, displaying how a document was altered/removed as well as estimated dataset level metrics (quantity of data removed in bytes or samples)). This rapid feedback loop enabled us to update the pipeline consequently in an iterative process to finetune each processing pipelines across datasets and languages with the input of native speakers. A specific example is shared in Appendix B.2. This resulted in 485 non-empty datasets.

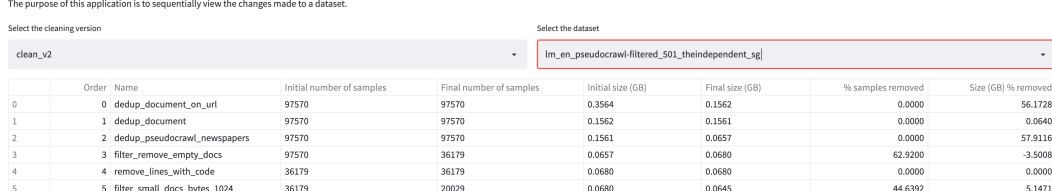

Figure 2: Partial screenshot of the visualization tool. Users can look at how each function in the processing pipeline influenced high-level statistics. Influence on specific samples can be monitored via the same tool, see Appendix B.2

## 3  Processing OSCAR

We chose to complement the data obtained at the end of the process described in the previous section with additional Common Crawl-based[8] data motivated by two main reasons. First, given the project's overall goal of providing a trained LLM as a research artifact comparable to previously released ones that have relied extensively on this source, we assessed that not including it would constitute too much of a departure and risk invalidating comparisons. Relatedly, recent work has put a strong emphasis on the quantity of data being a strong factor in a trained model's performance on evaluation tasks (Kaplan et al., 2020; Hoffmann et al., 2022), and we were missing about one third of data in order to optimize our compute budget in this direction. With that in mind, we chose OSCAR version 21.09 (Ortiz Suárez et al., 2020), based on the Common Crawl snapshot of February 2021, to make up the remaining 38% of our final dataset.

However, crawled data suffers from several known issues. First, we wanted to only select documents written by humans for humans, and exclude machine-generated content e.g. search engine optimization (SEO). Crawled content also over-represents pornographic text across languages (Kreutzer et al., 2022), especially in the form of spam ads. Finally, it contains personal information that may constitute a privacy risk. The present section outlines our approach to mitigating those issues.

### 3.1  Data cleaning and filtering

Our first approach to addressing the above consists in defining quality indicators for web content. These can then be used to filter out specific pages by defining cutoff thresholds. Extensive descriptions for reproduction are available in appendix C. We filtered out documents with:

- Too high **character repetition** or **word repetition** as a measure of repetitive content.
- Too high ratios of **special characters** to remove page code or crawling artifacts.
- Insufficient ratios of **closed class words** to filter out SEO pages.
- Too high ratios of **flagged words** to filter out pornographic spam. We asked contributors to tailor the word list in their language to this criterion (as opposed to generic terms related to sexuality) and to err on the side of high precision.
- Too high **perplexity** values to filter out non-natural language.
- Insufficient **number of words**, as LLM training requires extensive context sizes.

The languages that we eventually considered in OSCAR were the languages for which we were able to obtain hyperparameters and the cutoff values for each of these indicators by native speakers. Specifically, we considered Arabic, Basque, Bengali, Catalan, Chinese, English, French, Hindi, Indonesian, Portuguese, Spanish, Urdu, and Vietnamese. The code used for filtering OSCAR, along with the language-specific parameters and cutoff values, are publicly available. We then asked native speakers of each language to use our visualization tool[9] to establish the thresholds for each filter. The percentage of documents removed after applying all these filters is given in Table 1, and the percentage of documents discarded by each filter independently is given in 3.

---

[8]https://commoncrawl.org/
[9]https://hf.co/spaces/huggingface/text-data-filtering

| AR | EU | BN | CA | ZH | EN | FR | HI | ID | PT | UR | VI | ES |
|----|----|----|----|----|----|----|----|----|----|----|----|----|
| 20.3 | 5.2 | 48.8 | 21.1 | 23.1 | 17.2 | 17.0 | 25.7 | 10.4 | 12.6 | 15.8 | 21.3 | 16.9 |

Table 1: Percentage of documents removed by the filtering per language (ISO 639-1 code).

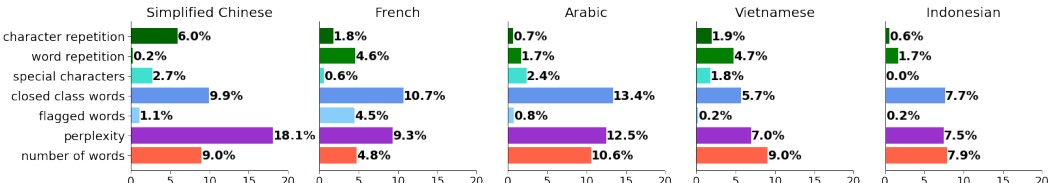

Figure 3: Percentage of documents discarded by each filter independently for 5 languages

## 3.2 Deduplication

Data deduplication has become a key tool for language model projects following research showing that it both improves performance on downstream tasks (Lee et al., 2022; Zhang et al., 2021) and decreases memorization of training data (Kandpal et al., 2022). To remove near duplicate documents in OSCAR (which is already exact-deduplicated) we initially used SimHash (Charikar, 2002; Manku et al., 2007), a hashing function that associates to two similar texts hashes with a low Hamming distance, with 6-grams and a Hamming distance threshold of 4. About 0.7% of the documents on average (0.07% ∼ 2.7%) were identified as near duplicates. However, because SimHash is essentially a bag-of-words algorithm, long documents are more likely to end up being similar to each other. In practice, we found false positives among long documents and decided not to discard documents in a same cluster of near-duplicates when they were longer than 6000 characters. Instead, we applied substring deduplication (Lee et al., 2022) based on Suffix Array (Manber and Myers, 1993) as a complementary method that clusters documents sharing a long substring, for documents with more than 6000 characters. We found on average 21.67% (10.61% ∼ 32.30%) of the data (in bytes) being duplicated.

## 3.3 Personally identifiable information

We used a rule-based approach leveraging regular expressions (Appendix C). The elements redacted were instances of *KEY* (numeric & alphanumeric identifiers such as phone numbers, credit card numbers, hexadecimal hashes and the like, while skipping instances of years and simple numbers), *EMAIL* (email addresses), *USER* (a social media handle) and *IP_ADDRESS* (an IPv4 or IPv6 address).

# 4 A First look at ROOTS

The efforts described in the previous sections come together in an assemblage of 1.6 Terabytes of multilingual text. Figure 4 puts that number into context by comparing the sizes of corpora typically used to train large language models. Documentation of the individual components of the corpus can be found in an interactive dataset card deck. In this section, we take initial steps towards further understanding of the corpus through statistical analyses of the aggregated data.

## 4.1 Natural Languages

The constitution of the corpus reflects the crowdsourcing efforts that enabled its creation. It comprises of 46 natural languages spanning 3 macroareas and 9 language families: Afro-Asiatic, Austro-Asiatic, Austronesian, Basque, Dravidian, Indo-European, Mande, Niger-Congo, Sino-Tibetan. At 30.03%, English constitutes the largest part of the corpus, followed by Simplified Chinese (16.16%), French (12.9%), Spanish (10.85%), Portuguese (4.91%) and Arabic (4.6%). A more detailed breakdown of the corpus can be found in the appendix and in an online interactive exploration tool[10],

---

[10]https://hf.co/spaces/bigscience-data/corpus-map

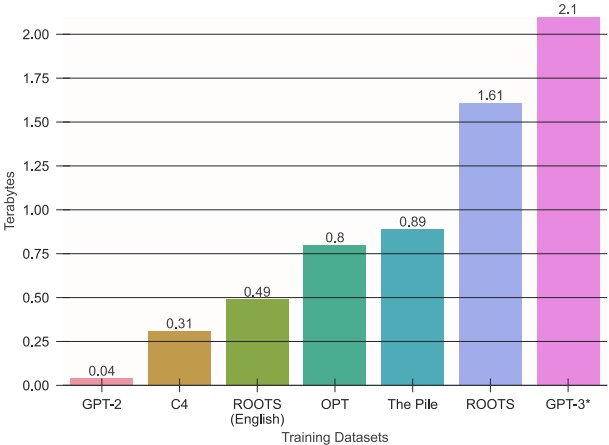

Figure 4: A raw size comparison to other corpora used to train large language models. The asterisk next to GPT-3 indicates the fact that the value in question is an estimate computed using the reported number of tokens and the average number of tokens per byte of text that the GPT-2 tokenizer produces on the `Pile-CC`, `Books3`, `OWT2`, and `Wiki-en` subsets of the Pile (Gao et al., 2020)

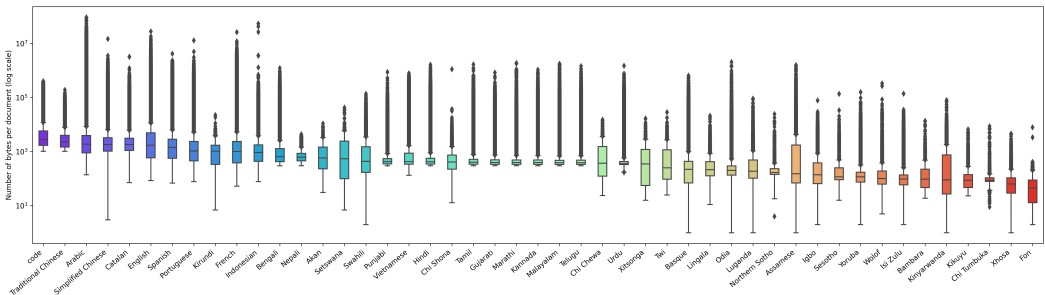

Figure 5: Size in bytes of every document in the corpus per language. The y-axis is in logarithmic scale. Box-and-whisker diagrams illustrate median, the first and third quartiles, whiskers drawn within the 1.5 IQR value and outliers

a screenshot of which is included in figure 1 to depict the byte-distribution of linguistic genera of the Eurasian macroarea subset of the corpus.

In order for the trained model to have an opportunity to learn long dependencies, the training corpus needs to contain long sequences of coherent text. At the same time, the previous post-processing steps only reduced the size of the documents. The median size of a document in our corpus is 1,129 bytes. Figure 5 shows the distribution of document sizes by language. A more detailed breakdown of the size of corpus on an online interactive tool.[11].

The distributions of the filter values for the different filters introduced in Section 3.1 and languages, for the Catalogue, Pseudo-Crawl and OSCAR (filtered) data are available in an online demo[12]. Examples for English are shown in figure 6. The different distributions reflect the diversity of sourcing and filtering of our main components. A notable example is the flagged word filter, for which the distribution for OSCAR is skewed right compared to the catalogue even after filtering.

## 4.2 Programming Languages

As depicted in the waffle plot in figure 1, the code subset of the corpus spans 13 programming languages, with Java, PHP, and C++ accounting for more than half of all documents.

---

[11] https://hf.co/spaces/bigscience-data/document-sizes
[12] https://hf.co/spaces/bigscience-catalogue-lm-data/filter_values_distributions

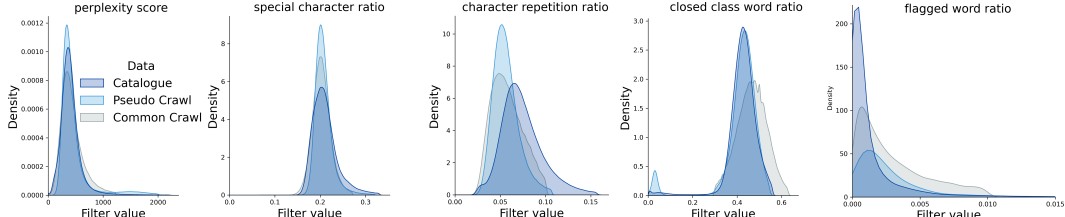

Figure 6: Some distributions of filter values for English. A filter value is the value that the filter gives to a document. These values are generally used to filter out documents that are too low or too high rated and also inform about the composition of the datasets.

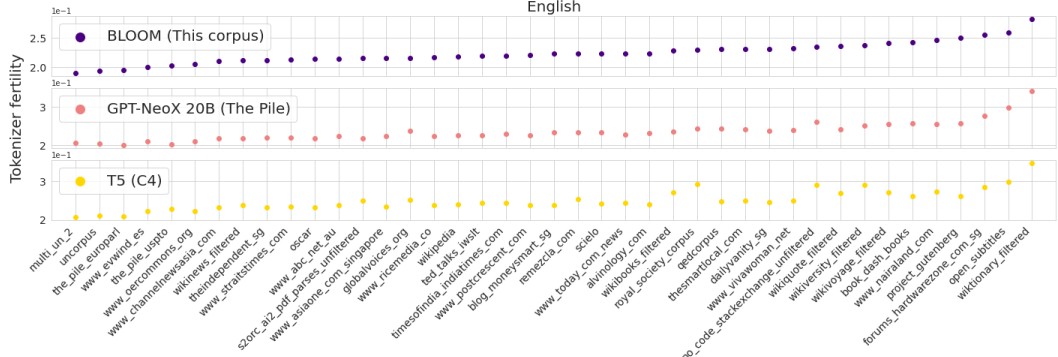

Figure 7: Tokens per byte for each English-language component for tokenizers trained on this corpus (BLOOM), the Pile (GPT-NeoX 20B) and C4 (T5). Lower values mean the component (X axis) is more similar in aggregate to the compared training corpus.

Configuration and test files are abundant in most GitHub repositories but not as interesting for code modeling. To that end, we use a heuristic whose first step examines the first 5 lines of a file for the presence of keywords such as "configuration file" or "test file". Failing that, the second step is to see whether the occurrence of the literals `config` and `test` in a given file exceeds 5% of the total number of lines of that file. We find that 5.23% of the data consists of configuration files and 7.88% of test files.

Allamanis (2019) and Lopes et al. (2017) highlight the large fraction of near-duplicates present in code datasets and how they can inflate performance metrics. Exact match deduplication alone can miss a fair amount of near-duplicates. To detect them, we first compute the MinHash of all documents, then create a Locality Sensitive Hashing (LSH) index between files to find the duplicate clusters in linear time. We additionally evaluate the Jaccard similarities within duplicate clusters to remove some false positives. We find 10.9M duplicate files in the clusters and 4.1M unique files: almost 32% of the data consists of near-duplicates. Syntax checkers[13] are used to validate 500K samples of Python and PHP code. We find that only 1% of the Python data and 2% of the PHP files do not pass the syntax check.

### 4.3 Tokenizer analysis of the component datasets

A tokenizer trained on a dataset can be used as a proxy for its content (Gao et al., 2020). The relevant metric is the number of tokens produced for a byte of natural language. The more different the training corpus from the tokenized corpus, the more tokens will be produced as the tokenizer is forced to divide natural text in more numerous, more general, smaller tokens. This property has allowed us to spot errors associated with outlier values, such as incorrectly classified languages, or crawling error. In the following analysis, we use it in two ways: first, we can use tokenizers trained on different corpora to see how ours differs from them; and second, we can use a tokenizer trained on this corpus to assess which components are outliers. We exclude outliers smaller than 5 documents.

---

[13]`py_compile` for Python and the `-l` flag for PHP

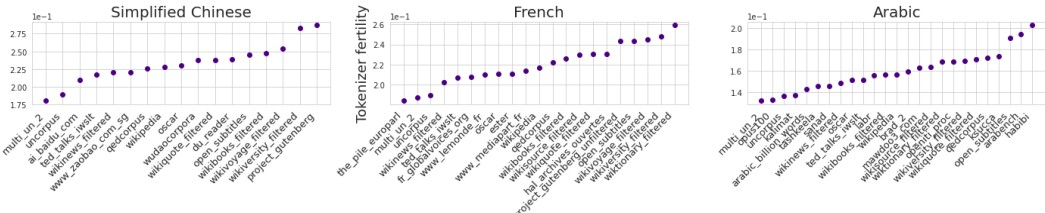

Figure 8: Tokens per byte for each French, Simplified Chinese, and Arabic component for tokenizers trained on this corpus. Lower values mean the component (X axis) is more similar in aggregate to the rest of the corpus.

Figure 7 shows the tokens-per-byte measurement on English component datasets for the BLOOM tokenizer, trained on this corpus, the GPT-NeoX 20B tokenizer (Black et al., 2022), trained on the Pile, and the T5 tokenizer (Raffel et al., 2020), trained on C4. Those tokenizers may differ in algorithms and/or vocabulary size, but we won't be directly comparing them to each other.

The figure is ordered by BLOOM tokenizer token-per-byte values, which shows that the ordering is very similar for BLOOM and GPT-NeoX. However, it shows several bumps for T5: component datasets that are out of domain in C4 but not our corpus, for example technical and academic datasets such as `s2orc` or `royal_society_corpus`, domains absent from C4's Common Crawl-sourced data. Other such datasets include `global_voices`, which contains news about non-English-speaking regions including quotes in the original languages and `no_code_stackexchange`, which contains forums which, although in English, may be dedicated to technical matters, foreign languages, or very specific domains. Both are similar to our corpus but not to the Pile or C4.

Figure 8 additionally shows BLOOM fertilities for Simplified Chinese, French and Arabic components. Outlier, high-fertility components, e.g. datasets that differ from the rest of our corpus, tend to be the same for all languages. `project_gutenberg` contains old books with their original formatting (for example, "**********" to denote page ends). `wiktionary` contains definitions of words in foreign languages. `wikiversity` contains technical terms and LaTeX. `wikivoyage` contains tables formatted as text. Forums may contain the user and date information of the message, as well as internet slang or emoji. `arabench` is spoken Arabic, and `habibi` is classical Arabic with more diacritics than modern. We deem most of those deviations acceptable to represent the diversity of uses of text, which tokenizer analysis is able to surface from the rest of the dataset.

## 5 Conclusion

We have presented ROOTS, a massive multilingual corpus that was the result of an international collaboration between multidisciplinary researchers studying large language models. The efforts to put the corpus together were value-driven and prompted by a data-first approach to training the BLOOM model. We further release the tooling developed throughout the project, and are currently implementing a release strategy that is informed by both the licensing and governance needs of every data source for the corpus itself. We hope this paves the way toward a more reflected use of the data that makes its way into large language models.

## Ethical Considerations and Broader Impacts Statement

As discussed in Section 1, the BigScience Research Workshop was conceived as a collaborative and value-driven endeavor from the start. This approach shaped many of the decisions described in this paper, spurring many contextual discussions and consensus-seeking on how to articulate the project's core values, those of the contributors to the data efforts, and considerations of social impact on the people directly and indirectly impacted. Of particular relevance were the data release and governance strategy, the choice to center human selection of data while still using OSCAR web-crawled for a significant section of the corpus, and the tools we developed to manage the risks of the latter (including regarding privacy). Each of these were the occasion of moral exercises and technical contributions that we believe were useful and required, and each will require further research and progress. We provide a more detailed discussion of these aspects of our work in Appendix A.

## Acknowledgements

**BigScience.** This work was pursued as part of the BigScience research workshop, an effort to collaboratively build a very large multilingual neural network language model and a very large multilingual text dataset. This effort gathered 1000+ reasearchers from 60 countries and from more than 250 institutions.

**Compute.** The BigScience Workshop was granted access to the HPC resources of the Institut du développement et des ressources en informatique scientifique (IDRIS) du Centre national de la recherche scientifique (CNRS) under the allocation 2021-A0101012475 made by Grand équipement national de calcul intensif (GENCI). Model training ran on the Jean-Zay cluster of IDRIS, and we thank the IDRIS team for their responsive support throughout the project, in particular Rémi Lacroix.

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
