# OpenReview forum: "The BigScience ROOTS Corpus: A 1.6TB Composite Multilingual Dataset"
_NeurIPS.cc/2022/Track/Datasets_and_Benchmarks — NeurIPS 2022 Datasets and Benchmarks _

### Official Review · Reviewer_EaMk · 2022-07-23
**The presented dataset at such a scale could benefit the community.**

**Rating:** 8
**Confidence:** 4
**Clarity:** The paper is clearly structured and w…

**Strengths:**

1. The paper presents a 1.6TB massive multilingual corpus as a result of 1-year international collaboration, containing curated data for 59 languages (46 natural languages and 13 programming languages), which could be used to support the training of large multilingual language models.

2. The paper releases the tools and codes of the processing pipeline used in the curation of the dataset, which could be beneficial to the community.

3. The paper presents a good documentation on the data sources and processing pipeline.

4. Native speakers are involved in the dataset curation, which could help improve the quality of the dataset.


**Weaknesses:**

1. The dataset covers only 46 natural languages and the motivation of choosing such languages is not well presented.

2. Regarding PII, although the paper presents muliwai as a filtering tool in the appendix, a rule-based approach was actually used. Additionally, the paper does not empirically evaluate the effectiveness of the rule-based approach in comparison to the more sophisticated muliwai.

3. It is also disappointing that the paper does not remove toxic texts with e.g., social bias, discrimination, etc.

4. The paper conducts deduplication on the two data sources (crowdsourced data and OSCAR) separately. However, it is not clear if it is necessary to deduplicate between the two data sources.

5. It would be very interesting to see how the data processing pipeline (with different settings) affect the finally trained LLM.

**Additional Feedback:**

When will the dataset be released?

**Correctness:**

The dataset is constructed as a result of international collaboration, where common operations, e.g., cleaning, filtering, deduplication, are performed at a large scale.

**Documentation:**

A website is used for the documentation of the dataset and tools.

**Ethics:**

It would be better if the paper could provide a discussion on issues related to PII or toxic data processing.

**Relation To Prior Work:**

It would be better if the paper could present a clear comparison to previous released datasets and tools.

**Summary And Contributions:**

The paper presents the 1.6TB multilingual dataset built for training the 176-billion-parameter BigScience large open-science open-access multilingual language model (BLOOM). The dataset contains both natural languages and programming languages, collected from two sources: crowdsourced datasets and OSCAR, a Common Crawl-based online data repository. Details of collecting, cleaning, merging, deduplicating on the two data sources are presented. In addition to the data processing pipeline, a preliminary analysis on the distribution over languages, document sizes, filter values and tokenizer analysis are conducted on the presented dataset.

The major contribution of this paper is to open source a large multilingual dataset with processing tools and details on data collection and processing to the community.

---

> ### Author Response · Authors · 2022-08-24
> **Curation choices and toxicity**
>
> We thank the reviewer for their feedback and will provide some clarification in this reply and in the final version of the paper.
>
> > The dataset covers only 46 natural languages and the motivation of choosing such languages is not well presented.
>
> We mention in the paper that “the language choice was chiefly driven by the communities who participated in the effort given the importance we placed on language expertise.” We endeavored throughout the paper to point out where language expertise was needed to drive curation choice to outline how much involvement from fluent speakers is required; the 46 languages are the ones for which we got interest and commitment from speakers. We will make this reasoning more explicit.
>
> > Regarding PII, although the paper presents muliwai as a filtering tool in the appendix, a rule-based approach was actually used. Additionally, the paper does not empirically evaluate the effectiveness of the rule-based approach in comparison to the more sophisticated muliwai.
>
> muliwai was not yet ready to apply at scale at the time of preprocessing and is part of an ongoing effort to develop better tools and benchmarks for PII driven by the scarcity of existing resources. We mention it in the spirit of transparency and to give an accurate picture of our processes including setbacks and negative results, and having to branch off out-of-scope work. We hope future efforts will be able to leverage it to increase the effectiveness of our PII mitigation approach.
>
> > It is also disappointing that the paper does not remove toxic texts with e.g., social bias, discrimination, etc.
>
> **This is an interesting point.** Our project was motivated in great part by ([Bender et al., 2021](https://dl.acm.org/doi/10.1145/3442188.3445922)) pointing out the ethical risks of training LLMs on filtered CommonCrawl dumps, especially in terms of over-representing hegemonic viewpoints and biases that lead to harms to communities and individuals. On the other hand, ([Dodge et al. 2022](https://aclanthology.org/2021.emnlp-main.98.pdf)) provide a cautionary tale for trying to automatically filter out and remove “toxic” content, and show that one person’s definition of toxic can translate into technology that harms already marginalized populations.
>
> This is what led us to choosing a “constructive” approach to creating a composite multilingual corpus made up of human-selected data sources for 62% of our data to mitigate those harms, while we outline our reasoning for including crawled data for the rest at the beginning of Section 3. We hope that this will enable further work on evaluating the impact of this alternative curation paradigm on a trained model and comparing the incidence of identifiable examples of various kinds of discriminatory text across both subparts of the corpus, so future large-scale curation efforts can further improve on our strategy.
>
> > The paper conducts deduplication on the two data sources (crowdsourced data and OSCAR) separately. However, it is not clear if it is necessary to deduplicate between the two data sources.
>
> Recent works seem to indicate that seeing items up to a handful of times makes little difference, so we were not too concerned about occasional duplication across the two halves ([Carlini et al. 2022](https://arxiv.org/abs/2202.07646)).
>
> > It would be very interesting to see how the data processing pipeline (with different settings) affect the finally trained LLM.
>
> We would also love to be able to compare the impact of different processing steps on the final model. However, given the significant compute budget required to train just one version of an LLM, testing each of them individually would be prohibitively expensive.

---

### Official Review · Reviewer_nTin · 2022-07-23
**Really multilingual corpus**

**Rating:** 7
**Confidence:** 4
**Clarity:** The paper is clear and well-written.

**Strengths:**

1. Big open collection of diverse texts should be useful for NLP community.
1. Datasets were chosen by native speakers.
1. Filtering of each language was done by a person proficient in this language.
1. I liked especially part where the authors investigate subset in terms of fertility. I think tokenization part is often overlooked in language modeling and this kind of analyzes shows, that the authors are aware of that.

**Weaknesses:**

1. I think part of programming languages should be another paper. They need a different approach for filtering and deduplication. I think they tend to have different set of biases (e.g. towards student assignments). In my opinion topic of creating corpus of code should be examined and described deeper, than it is possible in this work.
1. I'd like to see a size comparison to existing datasets.
1. If I understand correctly, the authors are not sure at the moment if data may be published. Without publishing the corpus this paper would be just an interesting analyze of internet content in terms of corpus creation.


**Additional Feedback:**

My score highly depends on sharing the data itself, and since at the moment there is no publicly available data, this is a truly borderline paper (can go both ways).

**Correctness:**

The paper as well as described pipeline reasonably constructed. Unfortunately, dataset usage isn't documented at the moment, but I hope it will change when the dataset will be published.

**Documentation:**

There is extensive documentation on the HuggingFaceHub of most of the content, but with regards to the issue I described in the `Correctness` section.

**Ethics:**

The authors themselves seems to have some issue with sharing the data, as far as I understand due to legal problems, so it's unclear to me.

**Relation To Prior Work:**

It is described clearly, however I'd like to see more statistics in comparison, at least dataset sizes.

**Summary And Contributions:**

The authors created a new, multilingual corpus for training large language models. They crowdsourced language corpora all around the globe and used them to create a new, big one. They will also share (as far as I'm concerned) part of it for further research.
This paper describes process of creation of the dataset, which contains crowdsourcing of corpora, pseudo-crawling, deduplication, filtering and cleaning. There is also provided analyze of the data.

---

> ### Author Response · Authors · 2022-08-24
> **Role and Reasoning for the Composite Data Release**
>
> We thank the reviewer for their detailed review. We also want to stress that while we are making different parts of the processed corpus available through different mechanisms (see [above](https://openreview.net/forum?id=UoEw6KigkUn&noteId=zip8ByQYXDZ)), the free-to-disseminate release of a single aggregated dataset was **not** the purpose of the work and is in fact contrary to its spirit. We provide further answers to the reviewer's comments:
>
>
> ## General Comments
>
> > I'd like to see a size comparison to existing datasets.
>
> We think that’s a great idea; a size comparison would indeed be useful and will be included in the final version of the paper.
>
> > I think part of programming languages should be another paper. They need a different approach for filtering and deduplication. I think they tend to have different set of biases (e.g. towards student assignments). In my opinion topic of creating corpus of code should be examined and described deeper, than it is possible in this work.
>
> The rationale behind the corpus was ultimately to train a large language model on both natural language and programming languages, which is why both had to be included. We agree that the code section itself is rather succinct, but that’s because the creation process itself is rather straightforward compared to natural languages. While the paper’s main focus is on creation rather than analysis, we do agree that a future full analysis of the different subsets will require different approaches, biases, and foci.
>
> > The paper as well as described pipeline reasonably constructed. Unfortunately, dataset usage isn't documented at the moment, but I hope it will change when the dataset will be published.
>
> At the time we submitted the current version of the paper, the [BLOOM family of language models](https://huggingface.co/models?other=bloom) was still a month away from being fully trained. The final version of the paper will reflect that.
>
> ## Data Availability
>
> > If I understand correctly, the authors are not sure at the moment if data may be published. Without publishing the corpus this paper would be just an interesting analyze of internet content in terms of corpus creation.
>
> Our main goal was to make the development of datasets of the size required to train LLMs more **open, collaborative, respectful of people’s rights** (including rights on data that they own or that they are represented in), and **reproducible**. To justify the need for such an effort, we point out that recent LLMs have either not made their training datasets available at all (with little detail on how they were made), released them in ways that go against the rights of the data subjects (including privacy, intellectual property rights, etc.), and/or made curation choices that go against the interests of significant parts of the population they aim to represent ([Dodge et al. 2022](https://aclanthology.org/2021.emnlp-main.98.pdf)).
>
> In order to create an LLM-sized multilingual dataset that does better than previous work on all these dimensions, we had to think out of the box by developing new tools, re-thinking the community involvement and organization of such an effort, and developing new ways of giving access to the data to promote research and transparency. The current work describes all of those contributions. First, for the sake of transparency and to enable analysis of models trained using the corpus in light of its curation process. And second, in order to enable future efforts to build on top of our work and re-use our tools, so we can keep making progress as a community on the very complex question of how to build a large-scale responsible dataset.
>
> For the first part, we take a composite approach to releasing subsets of the data so people can analyze it and use it for research as easily as possible. We have already identified 204 components that will be fully accessible upon agreeing to the project’s ethical charter, and we are inviting people who have an interest in analyzing the data and its impact on the BLOOM model to join the ongoing BigScience efforts so they can do so without disseminating it (which would be in violation of many of the data owners’ rights). For the second, we release all our tools and describe our methods and the difficulties we encountered.
>
> We hope that the reviewer will take another look at our submission in light of this context and evaluate its contributions beyond the (secondary) amount of processed data we release at the least restricted level of access. We are happy to answer further questions.
>
> > The authors themselves seems to have some issue with sharing the data, as far as I understand due to legal problems, so it's unclear to me.
>
> we hope that the answer above will have provided more clarity on this point. We additionally refer the reviewer to ([Jernite et al. 2022](https://dl.acm.org/doi/10.1145/3531146.3534637)) for further discussion of the reasoning behind and ethical stakes of data release and availability.

---

### Official Review · Reviewer_K4HT · 2022-07-24
**Review for The BigScience Corpus A 1.6TB Composite Multilingual Dataset**

**Rating:** 9
**Confidence:** 4

**Strengths:**

1. The development of this dataset is highly collaborative and open.
2. Many interactive tools are developed during the dataset construction, and could be helpful for future researches.
3. The data source selection considers many aspects, and helps in reducing biases in the proposed dataset.


**Weaknesses:**

I spot no major weakness in this work.

Minor:

1. The submitted main text does not have line numbers.
2. I can't help but notice that the number of data sources for China is only 5, and for Japan, it is 3.(https://huggingface.co/spaces/bigscience/SourcingCatalog)  Why is this imbalanced? I think there are a lot of internet users in China and Japan, hence more possible data sources, maybe there is still a bias towards the Eastern world?


**Additional Feedback:**

I would appreciate it if this work could involve more data sources on the Chinese and Japanese internet since Chinese and Japanese could also be the user of such a trained language model and there is indeed a very large web for Chinese and Japanese.

**Clarity:**

The paper is well written and easy to understand.


**Correctness:**

The construction of the dataset seems to be correct for me.


**Documentation:**

The dataset is very well documented on HuggingFace.


**Ethics:**

I believe the paper put in a lot of effort into addressing the ethical issues (which is the most important thing for this work), and I think foreseeable ethical issues are mostly addressed well.


**Relation To Prior Work:**

The paper discussed many related works and has a clear position in the literature.


**Summary And Contributions:**

This paper described an international effort of collecting a large-scale high-quality open dataset for language processing.
I believe this dataset could be of interest to a lot of people working on this topic and could be a solid ground for future research.
The effort of sourcing and filtering the corpus is detailed, many tools are used and developed, I think this could also help future works.
The issues of LICENSE of code and open access documents are considered which I think is the most critical factor in collecting such a huge dataset and training a model on it.
Overall, I believe this paper could be of interest to a lot of people (in fact there is already a lot of people who are interested in this work) and could be the cornerstone of future research, I would recommend strong accept for this work.

---

> ### Author Response · Authors · 2022-08-24
> **Cardinality of the Chinese catalog**
>
> We thank the reviewer for their feedback. We noticed the issue with the line numbers too late and corrected it for the appendix.
>
> As regards the number of Chinese and Japanese sources, we would like to first point out that only the former is part of the corpus' supported languages. We encouraged participants in our data sourcing effort to use our tools to index resources in whatever languages they were interested in, as we thought it would be a useful resource beyond the BLOOM model training, but ultimately we did not have enough participants to work on the other required steps to include Japanese as a supported language.
>
> As for Chinese, we did have more difficulty than for other languages given both our effort and the importance of the language. In particular, finding accessible sources of language data from mainland China proved a challenge. We note however that thanks to the inclusion of one very one resource (the WuDaoCorpora is by itself 200GB), Chinese is still one of the most represented languages by quantity in the corpus. In addition to the cardinality of the sources, a better representation of the linguistic makeup of the corpus can be found in Table 3 of the Appendix which shows the number of bytes of text for every single language present in the corpus, or in the following [interactive treemap](https://huggingface.co/spaces/bigscience-catalogue-lm-data/corpus-map).

---

### Official Review · Reviewer_nN25 · 2022-07-25
**This paper presents an initial subset of the corpus to train BLOOM and corresponding preprocessing methods and tools.**

**Rating:** 6
**Confidence:** 4
**Clarity:** This paper is well written.

**Strengths:**

* This paper presents BigScience Corpus which is used to train BLOOM.
* This paper releases preprocessing tools to produce BigScience.
* This paper gives a simple and first analysis at the BigScience Corpus.
* Details of how to produce this big corpus are given.
* This paper also gives a good example of collaboration between researchers around the world.
* The BigScience Corpus contains both document texts and code texts.
* The BigScience Corpus may have high influence on the NLP research community since it has both large scale clean data, released preprocessing tools and a trained BLOOM model.

**Weaknesses:**

* This paper lacks details of how to preprocess text data from various resources. More demo examples can help readers to understand how to deal with diverse text preprocessing problem.
* The "Personally identifiable information" section is too rough, which might cause privacy problem in the future. Since this corpus contains so much text data and code data.
* The analysis of this corpus is simple. Since this paper wants to show the power and help of big corpus, it would be more clear if the author show some trained model performance with different data sizes.
* one more concern is that it contains many existing NLP datasets (section 2.1) which might cause label leakage in following zero shot or few shot testing of pre-trained language model.

**Additional Feedback:**

N.A

**Correctness:**

* This paper present correct corpus preprocessing workflow and tools.
* This paper lacks elaborate data analysis based on trained model performance on this corpus.
* This paper lacks some ablation between multiple complex preprocessing steps. One pre-trained language model could produce different output with different training corpus. It is reasonable to provide some examples such as prompt results of trained model with/without preprocessing tricks provided in this paper.

**Documentation:**

The details or readme of preprocessing tools/reproduction could be better in its git repo.

**Ethics:**

The "Personally identifiable information" section is too rough, which might cause privacy problem in the future. Since this corpus contains so much text data and code data.

**Relation To Prior Work:**

This paper lacks elaborate comparison with current/existing large language model training data. This paper can provide the texts number of different language. It is also reasonable to give comparison results with current code generation data such as CodeXGLUE since it contains Code data.

**Summary And Contributions:**

This paper presents the BigScience Corpus which can be used to train large language model. Its contributions are two folds:
* With the help of international collaboration between multidisciplinary researchers, it provides a massive multilingual corpus (including text and code) which can be used to train BLOOM.
* The complex processing tools of developing this corpus are released, which may be helpful for future researchers to create/add more corpus data.
Although numerous tools and texts are presented in this paper, preprocessing examples and model performance trained on this corpus are  missing. Since this paper provides a large corpus to train large language model, it's reasonable to provide performance of trained small/big model for better insights.
Overall, the corpus presented in this paper is valuable and could be useful for other researchers.

---

> ### Author Response · Authors · 2022-08-24
> **Focus of the effort and required analysis**
>
> We thank the reviewer for their thorough reading of the paper.
>
> While we are as eager as the reviewer to see how models trained on the corpus behave on traditional NLP tasks, we want to stress again that our goal was to showcase how to gather a multilingual LLM-scale corpus **more responsibly by construction** rather than to optimize downstream performance. We further answer comments next:
>
> > This paper lacks details of how to preprocess text data from various resources. More demo examples can help readers to understand how to deal with diverse text preprocessing problem.
>
> We agree that an example is worth a thousand words! In addition to releasing the pre-processing code and describing the various steps applied to each of the sources, we also made [step-by-step exploration](https://hf.co/spaces/bigscience-catalogue-lm-data/process-pipeline-visualizer) available to show the exact impact of each of the filtering steps on specific examples from the catalog sources.
>
> > The "Personally identifiable information" section is too rough, which might cause privacy problem in the future. Since this corpus contains so much text data and code data.
>
> We did our best to limit the PII risks through a multi-pronged approach. First, we removed all PII we could reliably find, so our processed version of OSCAR has strictly less private information than the original (Appendix C). Second, we did our best to have enough data to train a model with a single pass and ran thorough deduplication, which decreases the risks of memorizing specific data points ([Carlini et al. 2022](https://arxiv.org/abs/2202.07646)). Third, since the efficacy of these methods is limited by the performance of personal information identification methods, we also release a work-in-progress tool to that end that will make future efforts like ours more efficient (Appendix C). We thought that describing the latter was beyond the scope of this paper but will describe the rest of our steps in more detail in the final version.
>
> > The analysis of this corpus is simple. Since this paper wants to show the power and help of big corpus, it would be more clear if the author show some trained model performance with different data sizes.
>
> We would love to hear more about what part of the papers suggested to the reviewer that the main goal of the effort was to improve technical model performance since that is not quite what we intended.
>
> Rather, we wanted to show what it takes to create a corpus in a way that is more open, community-driven, respectful of the rights of data and algorithm subjects, and collaborative than what has been done to date at that scale. Specifically, we were hoping to both provide transparency into the challenges we encountered, and enable future efforts that share similar aims by releasing tools and presenting positive and negative results together (including the limitations of our mitigation strategies). We will do our best to make this clearer and hope that the reviewer will find the paper is better aligned with those aims.
>
> > one more concern is that it contains many existing NLP datasets (section 2.1) which might cause label leakage in following zero shot or few shot testing of pre-trained language model.
>
> The corpus only contains unsupervised text without labels which reduces the risk of label leakage. Because language modeling requires document-level text rather than sentence-level text, most NLP datasets focused on sentence-level tasks were also excluded from the corpus from the start. However, some tasks can still be affected by the presence of corresponding text without labels in the training corpus (NLI, machine translation). For this reason, we've documented every dataset we've included to make it easy for researchers to know what's true zero-shot or not. We are also building a tool to index the common crawl-based parts of the corpus so that people can perform contamination analysis similar to that of ([Sanh et al., 2022](https://arxiv.org/pdf/2110.08207.pdf))
>
> > This paper lacks elaborate comparison with current/existing large language model training data.
>
> Most large language models of the scale of BLOOM (which was the driver for this corpus) do not publish their training data and resort to briefly describing it in the paper, which makes a meaningful comparison impossible.
>
> > This paper can provide the texts number of different language. It is also reasonable to give comparison results with current code generation data such as CodeXGLUE since it contains Code data.
>
> We agree with this remark and think it would be apt to include a comparison to the AlphaCode dataset, seeing as we used the language composition of that corpus to seed the creation of the programming subset of ours. Such a comparison will be added to the paper.
>
> > The details or readme of preprocessing tools/reproduction could be better in its git repo.
>
> You make a great point! We have made changes to the repository to make it easier to navigate and understand.

---

### Official Review · Reviewer_qpk1 · 2022-07-25
**BigScience corpus review**

**Rating:** 7
**Confidence:** 3
**Correctness:** I did not identify any statements tha…

**Strengths:**

* large, multilingual data set created in a principled manner and
  can be used for training better large language models (and not just for English)
* procedures for processing the data are clearly described
  and have interesting technical ideas (and the scripts were made
  publicly available)
* good comparison to other data sets using the tokenizer (4.3)


**Weaknesses:**

* it is not clear to which extent BigScience Corpus is about "science";
  judging by the name I'd expect a resource much more focused on science
  (interestingly, this type of focus is not clearly expressed in
  the abstract or the first paragraph of Introduction)
* the paper is lacking clear evidence for BigScience Corpus being
  superior to state-of-the-art data sets (Pile, C4, etc.), especially
  as far as the science aspect is concerned (the analysis based on tokenization
  is an exception, as I mentioned in Strengths)
* … in particular, is it really significantly more "science'y" than Pile, C4 etc.?
* … and not just in terms of raw statistics but as a more in-depth assessment
  (e.g. try to estimate whether you could "squeeze" more knowledge from this
  data set)


**Additional Feedback:**

* why such a large variance for percentage of documets filtered out (Table 1)?
  in particular why such a large percentage for Bengali?
  I think it should be briefly commented on

Minor fixes:

* "GPT3" -> "GPT-3"

* "collections of document" -> "collection of documents"

* "were furthered processed" -> "were further processed"

* "to avoid reduce replication" -> "to avoid reducing replication" (?)

* please fix capitalization of proper names in bibliography (e.g.
  ("What’s in the box? a preliminary analysis of undesirable
   content in the common crawl corpus."), see https://texfaq.org/FAQ-capbibtex

Minor comment:

* I'm curious why no Haskell among the programming languages? It is a
  programming languages different from other ones, closely related to
  mathematical ideas such as theory of category and with a substantial
  ecosystem of libraries, so in my opinion it would be good to have it
  in the Big*Science* corpus.


**Clarity:**

Quality of presentation is very good.

I think it would be good to have some specific, representative
examples of resources on which BigScience is based, given in the paper, e.g. in a table
(rather than in Appendix/supplementary materials/external resources).


**Documentation:**

The quality of documentation is definitely a strength of this paper.

**Ethics:**

The paper is lacking some discussion of copyright issues, including
source codes (especially in view of the fact that _only_ GPL-licensed codes
were incorporated, which is rather unfortunate, BSD/MIT/Apache licenses would
be less problematic).


**Relation To Prior Work:**

I assume this is the first paper introducing BigScience Corpus, right?
Maybe this should be made more clear? The BigScience Catalogue paper
(McMillan-Major et al. 2022) was cited, but maybe it should be
specified in a clearer manner what are the new contributions with
respect to that paper (and possibly other BigScience-related paper(s)?).


**Summary And Contributions:**

(I assume that this is the first paper in which BigScience as a large
multilingual set is introduced. See "Relation To Prior Work" where I express
some uncertainties).

The paper describes BigScience Corpus, a very large (1.6TB)
multilingual dataset. It is a very interesting initiative due to (1)
its multilinguality and (2) care taken as far as quality is concerned.

---

> ### Author Response · Authors · 2022-08-24
> **Naming and additional feedback**
>
> We thank the reviewer for their detailed comments. We provide clarification here for:
>
> ## Relation To Prior Work
>
> > I assume this is the first paper introducing BigScience Corpus, right? Maybe this should be made more clear? The BigScience Catalogue paper (McMillan-Major et al. 2022) was cited, but maybe it should be specified in a clearer manner what are the new contributions with respect to that paper (and possibly other BigScience-related paper(s)?).
>
> The elaboration of the BigScience corpus was a complex year-long endeavor with many different components. One of these components was the elaboration of a catalogue of language resources to be included in the corpus; that process is described in ([McMillan-Major et al. 2022](https://arxiv.org/abs/2201.10066)). The project however started before the catalogue with the elaboration of an overall strategy, continued after by obtaining and processing the data it pointed to, and ran in parallel including obtaining and processing data from a web crawl. We will make this relation clearer in the final version of the paper.
>
> ## The name BigScience vs. the content of the corpus
>
> > it is not clear to which extent BigScience Corpus is about "science"; judging by the name I'd expect a resource much more focused on science (interestingly, this type of focus is not clearly expressed in the abstract or the first paragraph of Introduction)
>
> We fully understand the confusion. Since submitting the paper, the project participants have decided on a name for the corpus which should help avoid its repetition in the future. The term BigScience is meant to qualify the research initiative that created the corpus. “Big Science” is a term that usually refers to large-scale open-science research projects (Such as the Large Hadron Collider at CERN) which the BigScience research initiative adopted as its founding inspiration. (See: https://bigscience.huggingface.co/.) The expression “BigScience corpus” is better understood as “The corpus created by the BigScience research collective”.
>
> ## Ethics
>
> > The paper is lacking some discussion of copyright issues, including source codes (especially in view of the fact that only GPL-licensed codes were incorporated, which is rather unfortunate, BSD/MIT/Apache licenses would be less problematic).
>
> The code subset of the corpus indeed only includes GPL-style licenses as a result of an incorrectly set boolean flag in the preprocessing pipeline. We meant to include all open licenses except GPL licenses. Since we won’t be releasing the entire corpus as a monolith, but rather in subsets each appropriately licensed, we decided it was acceptable to continue with that data and license this particular subset using a GPL-style license. All other data (source code or not) without a clear license from its creators will not be publicly available, but rather accessible to researchers according to our [data governance scheme](https://openreview.net/forum?id=UoEw6KigkUn&noteId=zip8ByQYXDZ).
>
> ## Additional feedback
>
> > why such a large variance for percentage of documents filtered out (Table 1)? in particular why such a large percentage for Bengali? I think it should be briefly commented on
>
> There is a large variance for each filter across languages (see Figure 3), hence also for the total number of documents filtered out. The thresholds for each filter were defined not with the objective of sticking to a predefined number, but to meet a need. Some languages needed to be filtered more than others, especially low-resource languages, for which the initial selection process for the source dataset was noisier on average to begin with (especially when it relied on automatic tools). Some languages, like Basque, did not need to be filtered at all by the flagged words filter, whereas it was more necessary for others (which echoes the findings of [previous multilingual OSCAR analysis](https://aclanthology.org/2022.tacl-1.4.pdf)).
>
> > I'm curious why no Haskell among the programming languages? It is a programming languages different from other ones, closely related to mathematical ideas such as theory of category and with a substantial ecosystem of libraries, so in my opinion it would be good to have it in the BigScience corpus.
>
> The choice of programming languages mirrors the choices taken by the DeepMind AlphaCode paper. The choice itself is driven by language popularity, and therefore the availability of large amounts of data that a large language model training corpus would necessitate. The main goal was to be able to eventually compare BLOOM’s code generation abilities with those of models trained solely on programming languages. We agree that including Haskell and other programming languages would be an interesting addition, and plan on exploring that in future work. Another point to keep in mind is the scarcity of Haskell data; indeed as per PyPL, Haskell's share of programs written is 0.3%.

---

### Official Review · Reviewer_Aj8f · 2022-07-28
**Significant step towards democratizing access to large language models**

**Rating:** 10
**Confidence:** 4

**Strengths:**

Documentation: Detailed documentation regarding data sources, processing and filtering applied. Code to reproduce publicly available.

Ethics: Value-first approach to data collection can inspire the broader research community to make this standard practice.

Significance: Datasets this size have traditionally not been published. The BigScience corpus and the BLOOM model trained using it are a significant step towards democratizing access to large language models.

**Weaknesses:**

Accessibility: dataset currently not publicly available, and authors stated intention to only publish subset of data. Would like for the authors to clarify the release strategy and governance.

**Additional Feedback:**

Clarifying questions for the authors:
1. Section 1: “release of a large subset of the BigScience corpus”. Why is only a subset of data to be released? Please clarify the release strategy and governance.
2. Section 2.1, Github code: “Due to a bug in the pre-processing pipeline the dataset was also filtered for GPL licenses only”. Does this mean that only code with GPL license was included in the dataset?
3. Section 2.1, Merging and Deduplicating sources: “We also removed datasets that we found had a high incidence of documents that were not fully in natural language”. What criteria are used for determining whether a string is natural language or not?
4. Was there any effort to exclude toxic language from the corpus? More broadly, what “censorship” has been applied?

**Clarity:**

Yes, with references to source code where appropriate leading to better reproducibility.

**Correctness:**

The paper describes the dataset collection and does not make any scientific claims.

**Documentation:**

Detailed description of data sources, processing and analysis of the resulting dataset. Code is open source.

**Ethics:**

See "Strengths" section.

**Relation To Prior Work:**

The authors claim that the open source approach to data collection stands in contrast to previous data collection efforts which have mostly been closed source.

**Summary And Contributions:**

The authors describe the construction of a 1.6TB multi-lingual text dataset purposely built for training large language models. The work has been value-driven and is inclusive of diverse language families and mindful of potential legal issues.

---

> ### Author Response · Authors · 2022-08-24
> **Clarification questions**
>
> We thank the reviewer for their comments and will try to include the following clarification points into the paper. For accessibility, we partly answer here and partly in the [comment above](https://openreview.net/forum?id=UoEw6KigkUn&noteId=zip8ByQYXDZ).
>
> > Section 1: “release of a large subset of the BigScience corpus”. Why is only a subset of data to be released? Please clarify the release strategy and governance.
>
> We wanted to create a high-quality multilingual dataset that respects the rights of its data creators and of its data subjects (including data owners). Limiting our choices to sources of text that can be freely disseminated without any ethical or legal issues would have introduced its own biases, and in particular, would have skewed toward even more English text. At the same time, BigScience was a research effort, so we wanted to make sure that future research could have access to the data even if it wasn’t directly downloadable.
>
> In order to respect both of these imperatives, we took a composite approach to release the data, with different processes for text coming from different sources. For some of them (204 sources x languages, including the Wikimedia-based ones), they can indeed be directly downloaded upon agreeing to respect the project’s ethical charter, while for others (such as the S2ORC corpus maintained by AI2), access to our processed version will depend on getting permission from the original data custodians. In addition, any researchers interested in studying the corpus or its impact on the model are welcome to join the ongoing BigScience efforts and study the text within the scope of its mission.
>
> > Section 2.1, Github code: “Due to a bug in the pre-processing pipeline the dataset was also filtered for GPL licenses only”. Does this mean that only code with GPL license was included in the dataset?
>
> Yes, the code subset only includes GPL-style licenses as a result of an incorrectly set boolean flag in the preprocessing pipeline.
>
> > Section 2.1, Merging and Deduplicating sources: “We also removed datasets that we found had a high incidence of documents that were not fully in natural language”. What criteria are used for determining whether a string is natural language or not?
>
> We considered a string of characters to be natural language if it is written by humans (not by machines) for humans (for example, not for search engine optimization (SEO)). We used various heuristics to approximate this criterion, including removing lines that do not have a high enough ratio of closed class words (see associated code [here](https://github.com/bigscience-workshop/data-preparation/blob/b56dd9e77456917f885eaaccbb39760d3c89c7a0/preprocessing/training/clean.py#L31-L34)) or lines that contain certain html tags (see associated code [here]( https://github.com/bigscience-workshop/data-preparation/blob/b56dd9e77456917f885eaaccbb39760d3c89c7a0/preprocessing/training/clean_helpers/map_remove_references.py#L5))
>
> > Was there any effort to exclude toxic language from the corpus? More broadly, what “censorship” has been applied?
>
> The corpus has two main parts that were filtered in different fashions. The “catalogue” subset of the corpus (62%) was made up of text sources that were individually selected by participants. We used some heuristics based on lengths and other criteria to remove examples that we thought were included by mistake but otherwise relied on the preprocessing undertaken upstream by the respective dataset authors and did not apply any topic-based filtering.
>
> The OSCAR subset of the corpus (38%) was obtained by filtering a dataset originally obtained by processing a CommonCrawl dump. We found an over-representation of pages that seemed to correspond to either pornography SEO or spam (including in unrelated websites). In order to remove those, participants compiled static lists of flagged words in their own languages (Bullet-point 4 in section 3.1 and appendix line 1048), and we removed documents that had a very high incidence of words from that list (we chose thresholds to prioritize accuracy, i.e. making sure that filtered documents were indeed mostly spam and SEO, over accuracy).

---

### Review · Ethics_Reviewer_1f5X · 2022-08-22

**Recommendation:** 1

**Ethics Documentation:**

Reviewers make explicit their rationale for data collection and organization and, more than that, the active steps taken to involve stakeholders in the creation and maintenance in the dataset. Far, far more care has gone into this project than most LLMs.

However, with so much care given to the creation of the dataset, it is perplexing that little to no space is given to consideration of ethical and responsible use of the dataset. The authors should consider the environmental, financial, and security risks posed by institutional use of the dataset (e.g., by law enforcement, by commercial software firms, etc). The Stochastic Parrots paper is cited as a model for dataset documentation, when, in reality, much of that paper focuses on dangers in LLM usage. Everyone has clearly gone into this project with the best intentions, but the authors should make clear what could or will happen when the dataset leaves its current community of practice and what steps could be taken to forestall anti-social uses.

**Ethics Review:**

This is an exemplary paper in its community-oriented approach to ML ethics. Authors clearly committed to doing BigScience with an explicit set of community-oriented values and evidence that throughout every stage of the dataset's creation pipeline. This reviewer was particularly impressed with the care given to complementary Common Crawl data.

---

> ### Author Response · Authors · 2022-08-24
> **Further thoughts on data governance for the corpus components**
>
> We very much appreciate the points raised by the reviewer!
>
> The overall BigScience efforts had several working groups focused on different aspects of working with data at this scale, including a dedicated group on data governance whose work was partly presented in [another publication](https://dl.acm.org/doi/10.1145/3531146.3534637), and a group on ethical and legal scholarship that explored both the normative and legal tools available for fostering positive uses of the data.
>
> We will add more references to the work of these groups to better reflect our consideration for these questions and the (necessarily incomplete) steps we took to mitigate these issues.

---

### Author Response · Authors · 2022-08-24
**Update on reviewer discussion and data availability**

We thank all reviewers for their comments and feedback on the paper. While we are still working on updating the paper to address some of these comments, we also wanted to already share some answers and provide some clarifications to the points raised to have some time for further discussion before the 29th.

First, as far as access to the data. We want to re-assert that the main goal of the paper was to provide a blueprint and supporting tools for responsible data curation with as much transparency as possible *rather than a fully free-to-disseminate aggregated dataset*. We are providing access to the process data to support this mission of transparency in the following way:
- 204 subsets out of 502 (one per source x language) are currently available upon agreeing to the project’s ethical charter at the same location as the exploration tools: https://hf.co/bigscience-catalogue-lm-data
- The data cards for each of these subsets can currently be accessed at: https://hf.co/spaces/bigscience-catalogue-lm-data/bigscience-corpus
- For the remaining 298 subsets, data governance efforts are ongoing to make them as available as possible while respecting the rights of their data subjects and intentions of their original custodians.

We're also encouraging people to keep joining the BigScience analysis efforts to be able to use the datasets strictly within the BigScience mission (i.e.: for the purpose or studying the data and its impact on the trained BLOOM model, but without disseminating it further). Anyone can join the effort, access is given to people working directly with the data.

We refer reviewers to individual answers to their comments for further information.

---

### Meta-Review · Area_Chair_FUmT · 2022-09-02

**Recommendation:** Accept
**Confidence:** 5

**Metareview:**

This paper describes the creation of the BigScience corpus used to train the BLOOM model. The paper describes steps taken to curate and construct the corpus as well as analysis of what it contains.

This is an impressive effort and blazes a trail among such data collection and governance efforts. Reviewers appreciate the democratization of a dataset of this size and generally appreciated the care taken in its construction, such as the filtering of data by native speakers of each language.  The authors convincingly rebut a few points about PII (criticisms which could be leveled against nearly any model at this scale, and which require whole lines of research to address in complete detail) and the aims of the corpus.

The most major point brought up is about the release process, particularly the decision to release a subset of the dataset. However, I am satisfied with the authors' responses, particularly to reviewer nTin.  I also agree with the ethics reviewer that significant care has been taken in this project, and although the data governance aspects of it are not discussed as much here, an accompanying publication describes them in detail. Taken together, these two papers are a model for how other efforts should proceed down the road.

Finally, in a similar vein, there are a few comments about different choices that could've been made (e.g., inclusion of source code, "toxic" content, etc.). However, unlike some NeurIPS papers, I don't think this project can be reasonably expected to jump through hoops for reviewers. The main question is: is the effort itself worthy of publication (yes), and is the documentation of the different parts of the effort clear and useful to the community in its present state (also yes). The decision-making process is laid out clearly (and elements from the rebuttal could be integrated to strengthen the paper further) and it seems clear that this effort is best-in-class in terms of transparency and other factors here.

---

### Decision · Program_Chairs · 2022-09-16

Accept